# Cancer/testis antigens FBXO39 and CEP55 expression correlates with survival in GBM patients

Parisa Azimi[1,2], Maryam Bazrgar[2], Taravat Yazdanian[3], Mehdi Totonchi[4*], Abolhassan Ahmadiani[2*]

**1** Department of neurosurgery, Alborz University of medical sciences, Karaj, Iran, **2** Neuroscience Research Center, Shahid Beheshti University of Medical Sciences, Tehran, Iran, **3** Neurological Clinical Research Institute and Healey and AMG Center for ALS, Massachusetts General Hospital, Harvard Medical School, Boston, Massachusetts, United States of America, **4** Faculty Member in Genetics and Stem Cell Departments, Royan Institute, Tehran, Iran

\* totonchimehdi@gmail.com (MT); aahmadiani@yahoo.com (AA)

## Abstract

### Background

Glioblastoma multiform (GBM) is a primary brain malignancy resistant to conventional therapies, with poor survival. Cancer testis antigens (CTAs) are important cancer diagnostic biomarkers and therapeutic targets. Bioinformatics analysis of GBM clinical and molecular data was undertaken to identify and validate the key CTA genes, whose expression correlates with the survival of GBM patients.

### Methods

RNA-seq data of GBM were downloaded from TCGA and CGGA databases to analyze differentially expressed genes (DEGs). Cancer Testis genes (from the CT database), found up and down-regulated in the GBM series compared to normal samples, were determined in both TCGA and CGGA databases. Overall survival (OS) of GBM patients in the TCGA and CGGA databases as a function of the expression of key CTA genes was considered to identify those whose expression significantly predicts patient survival. The predictive values of our candidate genes were then tested using our independent cohort (n = 29) using an RT-qPCR approach.

### Results

Compared to normal brain samples, a total of 2463 and, 6249 up- and 3706 and, 6606 down-regulated genes were found in the TCGA and CGGA, respectively. The intersection between 279 CTAs in the CD database and the up-and-down-regulated genes of both other databases was found in 11 and 8 CTAs, respectively. Using tumor samples from our cohort of 29 patients and an RT-qPCR approach, we found

**Data availability statement:** All relevant data are within the manuscript and its Supporting Information files.

**Funding:** The author(s) received no specific funding for this work.

**Competing interests:** The authors have declared that no competing interests exist.

**Abbreviations:** CNS, Central Nervous System; GBM, Glioblastoma Multiforme; HR, Hazard ratio; CGGA, Chinese Glioma Genome Atlas; TCGA, The Cancer Genome Atlas; OS, Overall Survival; WT, Wild Type; WHO, World Health Organization.

that FBXO39 and CEP55 genes were highly expressed in GBM and that their expression predicted the OS (P<0.05).

## Conclusions

Our results support the potential use of FBXO39 and CEP55 gene expression measurement as prognostic biomarkers in GBM patients.

## Introduction

Diffuse high-grade glioma tumors are the most aggressive brain tumors. The classification of brain tumors has undergone several changes over time due to various factors such as genetic variations. The World Health Organization (WHO) has played a key role in the effort [1]. According to the fifth edition of the WHO Classification of Tumors of the Central Nervous System (WHO CNS5), adult-type diffuse glioma grade IV with IDH wildtype called glioblastoma multiform (GBM) [1]. Glioblastomas accounted for 60.2% of all gliomas, 14.2% of all CNS tumors, and 50.9% of all malignant CNS tumors. Incidence of GBM increased slightly statistically significantly from 2000–2004 and 2004–2019. With an incidence rate of 3.27 per 100,000 persons in the United States, and more common in males, it is uncommon in children and increases with age [2]. The diverse factors such as incidence rates, tumor location, age of diagnosis, race, and the role of molecular genetics in the diagnosis of GBM, etc., are reported in various regions of the world. However, making comparisons of some data, e.g., GBM incidence was complex due to significant differences in the methodology of epidemiological reports [3]. The GBM is the most common lethal brain tumor with poor treatment outcome, despite the best therapeutic regimen [4]. The rate of local recurrence of GBM is high with a 5-year overall survival (OS) rate of less than 5% [5]. They are described by extensive intra- and inter-tumoral heterogeneity at molecular and cellular levels [6]. Therefore, it is important to seek specific predictive and prognostic molecular signatures for individualized targeted therapy [7].

Cancer testis antigens/genes (CTA/CTG) are described as a group of proteins/ genes that are preferentially expressed in normal male germ cells, gametes, and trophoblasts while silenced in most somatic tissues [8]. They are abnormally expressed in about 40% of various types of human cancers and could provoke spontaneous humoral and cell-mediated immune responses in cancer patients [9]. These characteristics make them appropriate prognosis markers as well as promising targets for innovative anti-cancer therapies [9]. To date, an estimated 279 proteins related to the CTA family have been documented [8]. However, the role of CTA gene expression as a diagnostic, and predictive marker of GBM tumors has not been fully studied. To better understand the prognostic potential of CTAs in GBM tumors, a thorough analysis of GBM-associated CTA gene expression in the open databases is required. In addition, it is also possible that these specific antigens may be potential targets for immunotherapy in future clinical trials and pre-clinical research.

To consider CTA gene expression as a potential prognostic marker in GBM tumors, we performed the analysis of the expression of selected GBM-associated CTAs and assessed their impact on the survival of GBM patients from public databases. The results have been validated with samples of our own patients' cohort.

## Methods

### Data processing

The RNA-seq and relevant clinical data of GBM patients, and normal controls were acquired from The Cancer Genome Atlas (TCGA, https://www.cancer.gov/) and the Chinese Glioma Genome Atlas (CGGA; http://www.cgga.org.cn/) database (CGGA.mRNAseq_693) [10]. A total of 418 GBM and 25 normal brain samples, including 169 and 5 from TCGA and 249 and 20 from the CGGA were included in the present study. Demographic information of two independent GBM datasets is shown in Table 1.

**Table 1. Demographic information of two independent GBM datasets.**

| Demographic categories | CGGA (n = 249) | TCGA(n = 169) |
|---|---|---|
| **Age** | | |
| Over 60 | 67 | 89 |
| Under 60 | 182 | 80 |
| **Gender** | | |
| Female | 102 | 59 |
| Male | 147 | 110 |
| **IDH-status** | | |
| IDH1-Mut | 49 | 11 |
| IDH1-WT | 190 | 141 |
| NA | 10 | 17 |
| **Subtype** | | |
| Primary | 140 | – |
| Recurrent | 109 | – |
| Classical | – | 39 |
| Mesenchymal | – | 50 |
| Neural | – | 26 |
| Pro-neural | – | 30 |
| G-CIMP | – | 8 |
| NA | – | 16 |
| **Radiotherapy** | | |
| Treated | 195 | – |
| Un-treated | 34 | – |
| NA | 20 | – |
| **Chemotherapy** | | |
| Treated | 201 | – |
| Un-treated | 29 | – |
| NA | 19 | – |

NA, not available; G-CIMP, cytosine-phosphate-guanine (CpG) island methylator phenotype (G-CIMP).

## Screening differentially expressed genes (DEGs)

The raw data of transcriptome profiles in highly expressed and lowly expressed GBM patients were obtained from the TCGA and CGGA datasets. To screen the DEGs between GBM and normal samples, the R package "limma" was applied. p-value < 0.05 and |log2 (fold change) | > 1 were regarded as the screening thresholds for DEGs.

## Identification of CTAs in GBM

The list of CTAs was obtained in the Cancer Testis database (CTdatabase) (http://www.cta.lncc.br/). The intersection between 279 CTAs in the CD database and the up and down-regulated genes of the TCGA and the CGGA datasets were identified separately using the Venn diagram Tool (https://bioinformatics.psb.ugent.be/webtools/Venn/).

## Overall survival analysis

After intersection analysis, CTA genes were obtained in GBM. Overall survival (OS) was analyzed using the GEPIA2 (http://gepia2.cancer-pku.cn/#survival) and the CGGA databases (http://www.cgga.org.cn/analyse/RNA-data.jsp) with a threshold of log-rank P < 0.05.

## Patients and tissue samples in our cohort

This is a prospective study. From February 2020 to January 2023, 45 adults were operated for primary GBM at our Institute. Histologic classification of brain tumor samples using the current WHO CNS5 classification identified 29 glioblastomas, which were included in the study. Two normal brain tissue samples were used as controls in molecular analyses.

Tumor and normal tissue samples were immediately frozen and stored at −80°C until use. For expression and survival analysis, we employed the median gene value to categorize tumor samples into high and low-expression groups. Subsequently, the expression data were merged with clinical survival data using sample barcodes. GraphPad Prism was employed to model these two groups' survival time and status. Log-rank test was utilized for the comparison of Survival Curves.

## Ethics statement

All patients provided written consent to use their specimens for research purposes, none of them was identifiable. The study was approved by the Ethics Committee of the Shahid Beheshti University of Medical Sciences (Code: IR.SBMU. REC.1398.023, Tehran, Iran). The ethical principles of the Declaration of Helsinki were respected by researchers in all stages of the research.

## RNA extraction and quantitative RT-PCR

RNA was extracted from frozen samples using the AnaCell RNA extraction kit and its quality and quantity were measured with a NanoDrop 2000c Spectrophotometer. Two micrograms of RNA were used for cDNA synthesis with the AnaCell cDNA synthesis kit. Quantitative reverse transcriptase PCR (qRT-PCR) was performed using SYBR Green and an ABI RT-PCR machine. GAPDH was selected as a reference gene. The $2^{-\Delta\Delta Ct}$ method was applied to tackle all the data.

Primer sequences were as follows: F-Box Protein 39 (FBXO39), forward, 5′- CCTGCGTGTATTCAAGGCGAG-3′; reverse, 5′-AAGTGCTCCCAGCTTTCTTCA-3′; Centrosomal Protein 55 (CEP55), forward, 5′- CAGGGAGGGCAGACC ATTT-3′; reverse, 5′- GGCTTCGATCCCCACTTACT-3′; GAPDH, forward, 5′-TGAAGGTCGGAGTCAACGGATTTGGT-3′, reverse, 5′-CATGTGGGCCATGAGGTCCACCAC-3′.

## Cox regression analysis of factors contributing to overall survival of GBM patients

Overall survival was defined as the time from the start of treatment to death or last follow-up based on public databases. This overall survival time was the outcome of interest. Multivariate Cox regression included 11 gene expressions, age,

gender, and primary/recurrence status of GBM patients. Samples from the TCGA and CGGA databases, separately, were used. Hazard ratios (HRs) and 95% confidence intervals (CIs) were calculated accordingly. Then, factors with significant prognostic values were assessed.

## Statistical analysis

Statistical analysis was carried out using GraphPad Prism version 9.2.0 (GraphPad Software, San Diego, CA). The multivariate Cox regression analyses were employed to recognize the independent clinical prognostic factors using the Cox regression in SPSS. The Mann–Whitney test was conducted to compare the differences in the expression of the examined markers in the GBM tissues and normal tissues. Kaplan-Meier curves and the log-rank test were used to assess the survival of patients with GBM. FBXO39 and CEP55 gene expression data were split into two groups for analysis: "high expression" and "low expression" by the median as the cut-off value. $P < 0.05$ was considered to indicate a statistically significant difference.

## Results

### Identification of DEGs in GBM versus control sample

RNA-seq data analysis from the TCGA database demonstrated that 2463 genes were significantly overexpressed and 3706 were significantly lower expressed in GBM tissues compared to normal samples (S1 File). Based on the RNA-seq data analysis from the CGGA database between GBM and normal samples, we obtained 6606 downregulated DEGs and 6249 upregulated DEGs (S2 File).

### CTAs found in GBM

As Fig 1a shows, 11 CTA encoding genes are up-regulated in GBM compared to control, which is an intersection between 279 CTAs, and the up-regulated genes of both TCGA and CGGA databases. All these genes include CTCFL, PLAC1, FBXO39, LEMD1, PRAME, POTEG, CEP55, DKKL1, PBK, ATAD2, and TTK.

   The intersection between 279 CTA and the down-regulated genes of both TCGA and CGGA databases showed that 8 CTAs are decreased in GBM compared to healthy people. This gene list includes SPAG6, CABYR, GPAT2, ELOVL4, TMEFF1, TMEFF2, MAGEC3, and SYCE1 (Fig 1b). (S3 File). The expression levels of FBXO39 and CEP55 were significantly up-regulated in GBM tissues compared to normal tissues and were visualized using the TCGA database, shown in Fig 2a and 2b, respectively. Similar results were observed in the CGGA database analysis.

### Evaluation of overall survival for CTAs

TCGA and CGGA databases were used to assess the association of up-regulated and down-regulated CTAs with overall survival (OS) in GBM. Notably, the GEPIA database showed that only higher expression of FBXO39 has a significant correlation with poorer OS (Fig 3a; p-value=0.013). Also, the CGGA database verified a correlation of higher expression CEP55 with poorer OS (Fig 3b; p-value=0.025). However, no significant effect was indicated for the remaining down-regulated CTAs regarding OS.

   As shown in Fig 4, genes (CTCFL, PLAC1, LEMD1, PRAME, CEP55, DKKL1, PBK, ATAD2, and TTK) from the GEPIA2 dataset, and genes (PLAC1, FBXO39, LEMD1, DKKL1, PBK, ATAD2, and TTK) of primary and recurrent GBM from the mRNA_seq693 of the CCGA cohort were not associated significantly with OS in GBM patients. For gene (POTEGE), and genes (CTCFL, PRAME, and POTEG), Kaplan-Meier was not analyzed due to the small sample size in the GEPIA2 and CGGA datasets, respectively.

### Gene expression and overall survival analysis in our cohort

To validate our finding on the prognostic value of FBXO39 and CEP55 gene expression, we prepared RNA from our tumor samples of 29 patients. A summary of clinical information of our patient cohort can be found in Table 2. We developed an

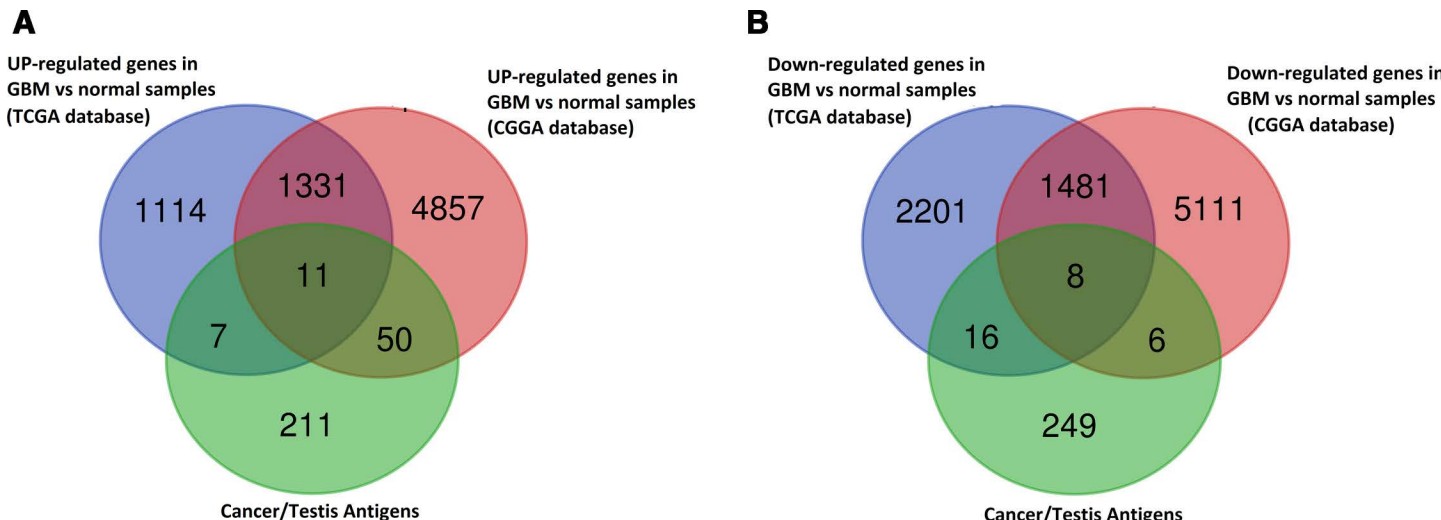

**Fig 1. Summary of identified CTA involved in the high expression.** Different colors of the Venn diagram represented different analysis results, **A**: green: 279 CTAs in the CT Database; blue: 2463 genes significantly up-regulated in GBM vs control in the TCGA database; red: 6249 genes significantly up-regulated in GBM vs control in the CGGA database; A total of 11 CTAs was obtained from the intersection set; **B**: green: 279 CTAs in the CT Database; blue: 3706 genes significantly down-regulated in GBM vs control in the TCGA database; red: 6606 genes significantly down-regulated in GBM vs control in the CGGA database; A total of 8 CTAs was obtained from the intersection set.

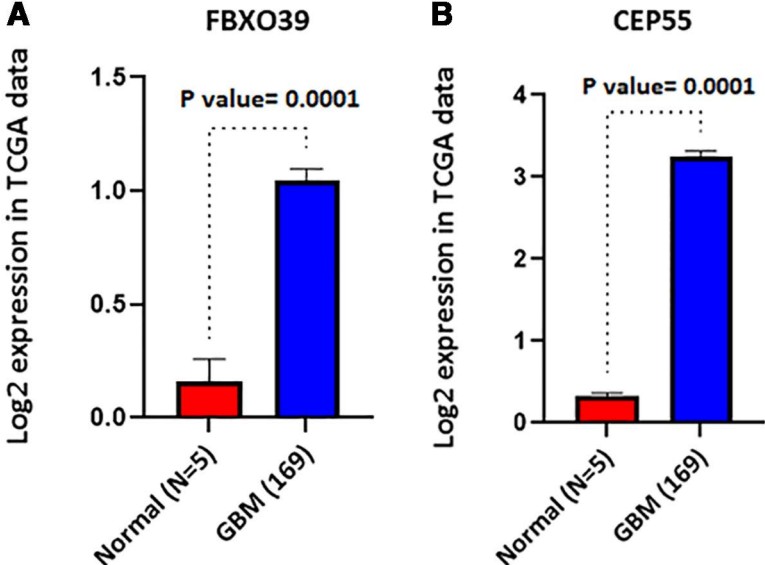

**Fig 2. Data from TCGA showed the high expression level of FBXO39 and CEP55 in GBM tissues and normal tissues, shown in 2a, and 2b, respectively.**

RT-qPCR test to examine the expression levels of FBXO39 and CEP55 genes in our cohort of 29 GBM patients and 2 cases of normal brain tissues. The expressions of FBXO39 and CEP55 were significantly up-regulated in GBM tissues compared to normal tissues (Fig 5a and 5b; S4 File). We further analyzed the power of FBXO39 and CEP55 in the prognosis of GBM using our cohort and found the existence of a trend toward worse overall survival in GBM patients with high expression of FBXO39

**A**

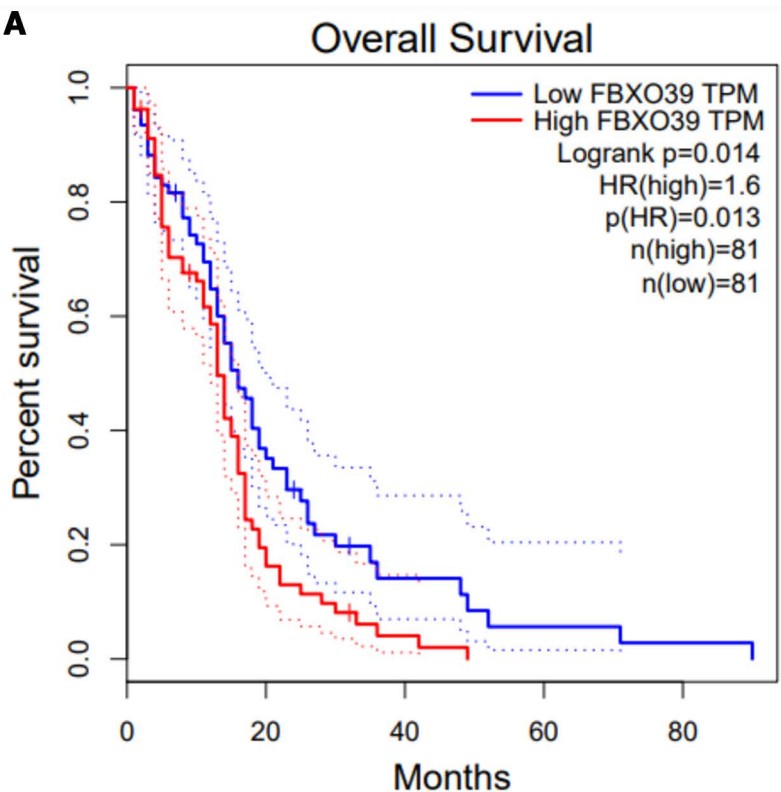

**B**

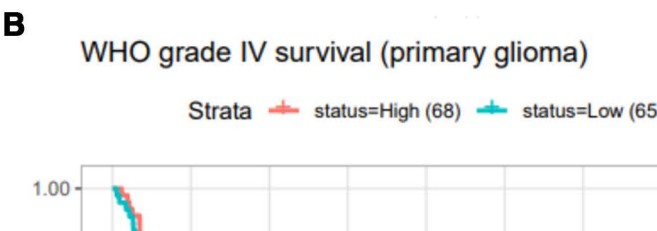

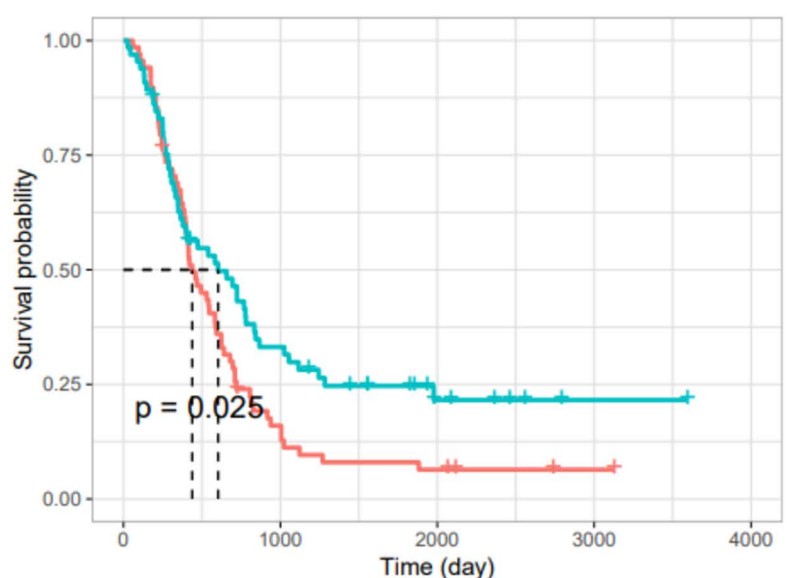

**Fig 3. Data from the Kaplan–Meier plotter showed the OS of GBM patients, a. stratified by expression of FBXO39 (from GEPIA) and, b. CEP55 (from CGGA).**

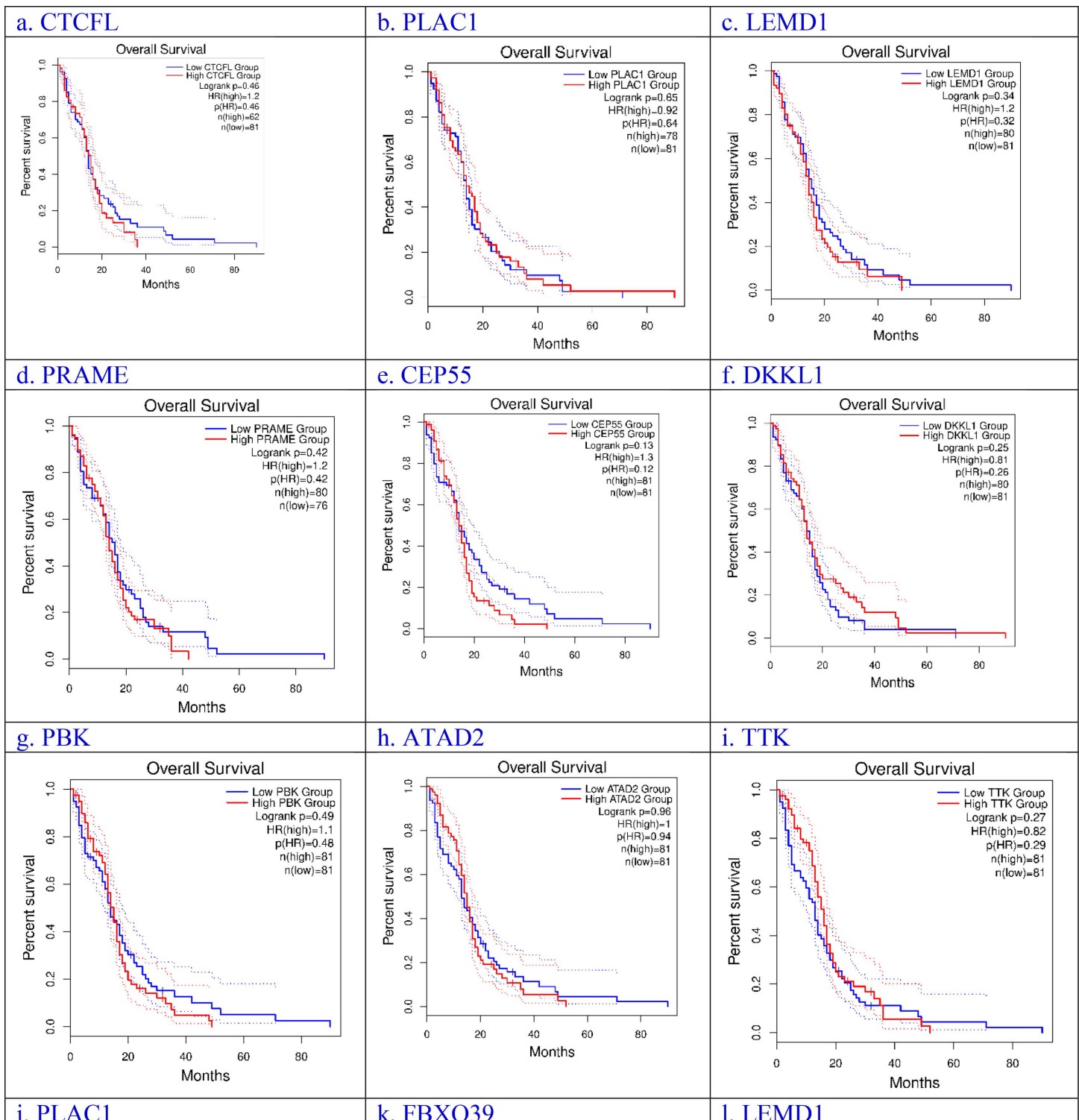

**Fig 4. (a–p) Kaplan-Meier analysis of overall survival for GBM patients in the GEPIA2 using the TCGA cohort (a. CTCFL, b. PLAC1, c. LEMD1, d. PRAME, e. CEP55, f. DKKL1, g. PBK, h. ATAD2, and i. TTK), and primary GBM from the mRNA_seq693 of the CCGA (j. PLAC1, k. FBXO39, l. LEMD1, m. DKKL1, n. PBK, o. ATAD2, and p. TTK) based on low- and high-expression of genes.** The red line represents samples with high expression of the genes, and the blue line represents the samples with low expression of genes. Among 11 genes, p<0.05 was considered to be statistically different.

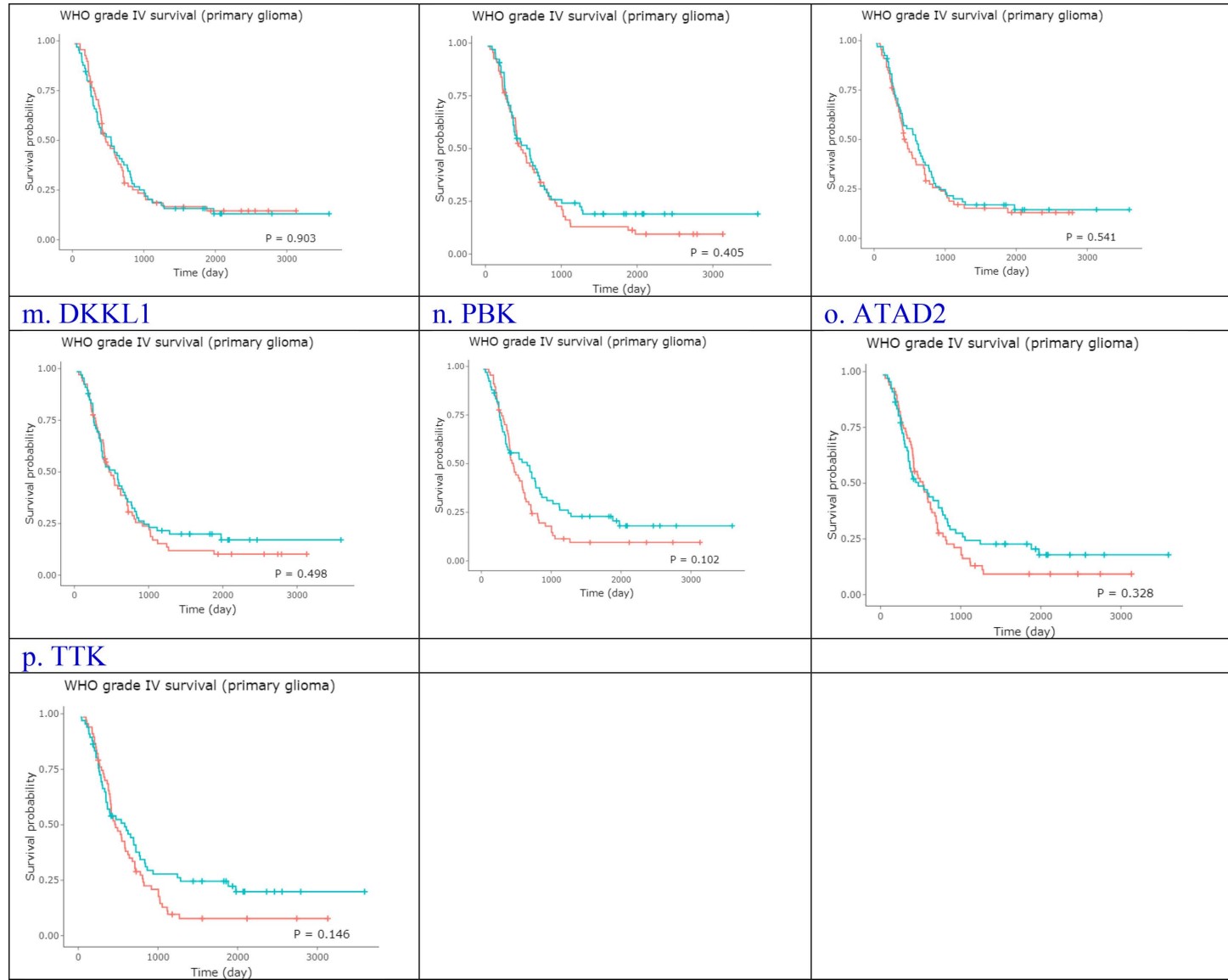

**Fig 4.** Continued.

(Fig 6a; S4 File). The same analysis was executed for CEP55 gene expression, and we found that similarly to FBXO39, there is a correlation between high CEP55 expression levels and overall survival of GBM patients (Fig 6b; S4 File).

The multivariate Cox regression analysis was performed to obtain the independent prognostic feature factors. As shown in Tables 3 and 4 (S5 and S6 Files), factors were found to be independently correlated to OS prognosis in the TCGA and the CGGA datasets, respectively. However, only, the LEMD1 gene and primary/recurrence status were significant in the CGAA cohort.

## Discussion

Despite efforts to improve the diagnosis and therapy in GBM patients, the 5-year survival rate and prognosis of these patients remains poor. An enhanced understanding of GBM progression is needed for the development of novel therapies.

**Table 2. Glioblastoma patient cohort characteristics (n = 29).**

| Characteristics | n (%) |
|---|---|
| **Gender** | |
| Male | 18(56.3) |
| Female | 11(34.4) |
| **Age in diagnosis** | |
| Mean (years) (SD) | 47.6 (15.5) |
| Range (years) | 22-75 |
| **KPS (preoperative)** | |
| 100% | 12 (41.4) |
| 90% | 7 (24.1) |
| 80% | 5 (17.2) |
| 70% | 3 (10.3) |
| <70% | 2 (6.9) |
| **Overall Survival** | |
| Mean (months) (SD) | 14.6 (6.5) |
| Range (months) | 0.5 - 24 |
| **IDH** | |
| Wild type | 22 (75.9) |
| Mutant type | 7 (24.1) |
| **EOR** | |
| GTR | 27 (93.1) |
| STR | 2 (6.9) |
| **Adjuvant CT** | |
| TMZ | 29 (100) |
| ACNU | 0 |
| BCNU | 0 |
| **Adjuvant RT** | 29 (100) |
| **Alive in the last follow-up** | 13 (40.6) |

EOR, extent of resection; GTR, gross total resection; KPS, Karnofsky performance scale; IDH, Isocitrate Dehydrogenase; RT, radiotherapy; STR, subtotal resection; TMZ, temozolomide; ACNU, nimustine; BCNU, carmustine; CT, Chemotherapy; NA, not available.

In this study, we identified a significant increase in FBXO39 and CEP55 expression in GBM tumor tissues compared to normal brains by wet bench as well as by bioinformatic pubic data analyses. Specifically, high levels of GBM tumor-specific expression of these genes correlated with poor survival. Therefore, our findings suggest that FBXO39 and CEP55 may act as an oncogenic gene in GBM's progression. These genes could be therefore considered as potential novel targets for the therapy of GBM patients and the understanding of their function could indicate possible new directions for GBM tumor research. Moreover, indeed, by nature, the CTA-encoding genes should be found only within the genes that are activated in cancer, since by definition, they are only expressed in the testis and not in normal somatic tissues. However, the brain is a particular tissue concerning the CTA encoding gene expression, since many testis-specific genes are known to be also expressed in the brain. We have also considered the list of significantly down-regulated genes compared to normal brain samples and checked if any of them is present in the list of CTA-encoding genes. We then considered the association between their down-regulation and the patient's survival. However, no statistically significant effect was observed on down-regulated CTAs regarding OS, albeit, more studies are needed to confirm the findings.

Published research showed that FBXO39 cancer/testis antigen was considered a promising biomarker for the prognosis of cervical squamous cell carcinoma [11], breast [12], colon [13], and glioma cancer [14]. FBXO39 plays vital roles

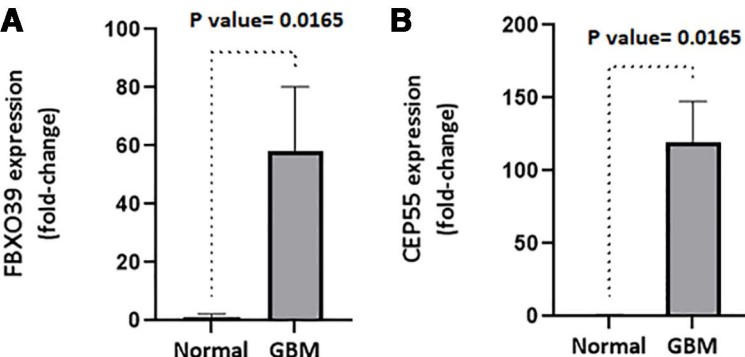

**Fig 5. a. and b. The expressions of FBXO39 and CEP55 were significantly up-regulated in 29 GBM tissues compared to 2 normal tissues, respectively.**

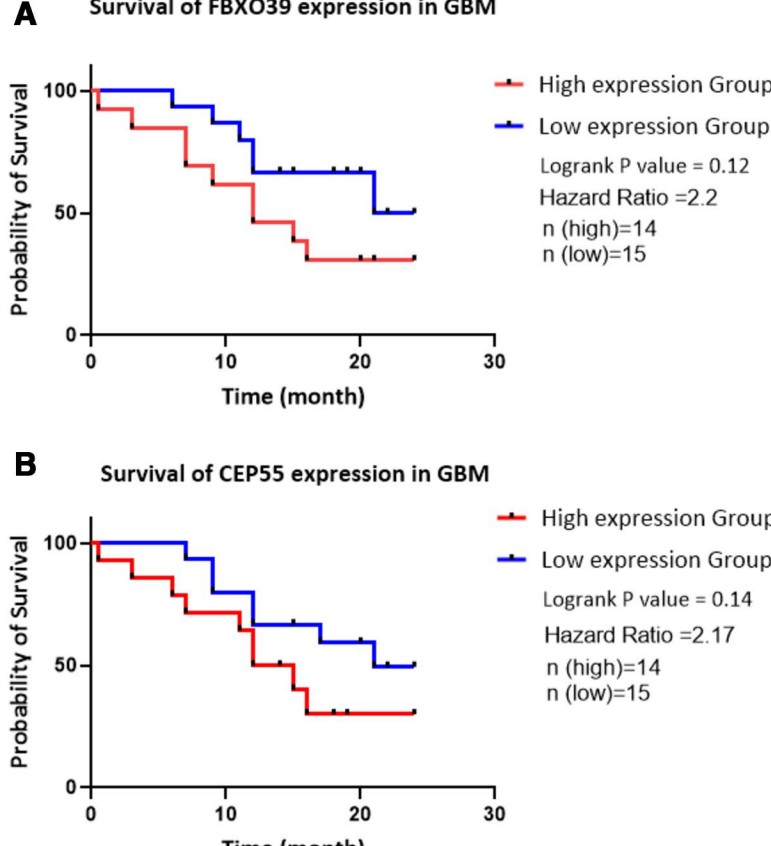

**Fig 6. a. and b. Kaplan-Meier plots of overall survival of GBM patients, stratified by expression of FBXO39 and CEP55, respectively.**

in cell cycle regulation, transcriptional regulation, apoptosis, and cell signaling. Wu et al. first presented that FBXO39 is up-regulated in glioma tissue. They documented that overexpression of FBXO39 is positively correlated with the diagnostic grade of glioma and negatively associated with patient survival rate [14]. In agreement with the previous study, our

**Table 3. Cox analysis from TCGA database. The list of independent prognostic feature factors.**

| Factor | β | P value | Hazard Ratio | 95%CI |
|---|---|---|---|---|
| FBXO39 | −0.354 | 0.554 | 0.702 | 0.218-2.263 |
| CTCFL | −6.913 | 0.596 | 0.001 | 0.000-122979137.2 |
| CEP55 | 0.185 | 0.095 | 1.203 | 0.968- 1.494 |
| PLAC1 | −0.006 | 0.980 | 0.994 | 0.619- 1.597 |
| LEMD1 | 0.075 | 0.303 | 1.078 | 0.934-1.245 |
| PRAME | 0.028 | 0.122 | 1.029 | 0.992-1.067 |
| POTEG | 2.665 | 0.705 | 14.365 | 0.000- 14038910.3 |
| DKKL1 | −0.126 | 0.741 | 0.881 | 0.417-1.861 |
| PBK | −0.018 | 0.584 | 0.982 | 0.920-1.048 |
| ATAD2 | 0.119 | 0.257 | 1.126 | 0.917-1.382 |
| TTK | 0.016 | 0.935 | 1.016 | 0.688-1.501 |
| Age | 0.003 | 0.851 | 1.003 | 0.975-1.032 |
| Gender | 0.095 | 0.808 | 1.099 | 0.511-2.367 |
| Primary/Recurrence | 0.672 | 0.212 | 1.959 | 0.682-5.624 |

**Table 4. Cox analysis from CGGA database. The list of independent prognostic feature factors.**

| Factor | β | P value | Hazard Ratio | 95%CI |
|---|---|---|---|---|
| FBXO39 | 0.196 | 0.109 | 1.217 | 0.957-1.547 |
| CEP55 | −0.166 | 0.656 | 0.847 | 0.408-1.759 |
| PLAC1 | 0.631 | 0.407 | 1.880 | 0.422-8.374 |
| LEMD1 | 0.178 | **0.041** | 1.194 | 1.007-1.417 |
| PRAME | 0.002 | 0.958 | 1.002 | 0.928-1.082 |
| DKKL1 | −0.053 | 0.443 | 0.948 | 0.827-1.087 |
| PBK | 0.047 | 0.244 | 1.048 | 0.969-1.133 |
| ATAD2 | −0.115 | 0.302 | 0.891 | 0.716-1.109 |
| TTK | −0.319 | 0.178 | 0.727 | 0.457-1.157 |
| Age | −0.015 | 0.313 | 0.985 | 0.957-1.014 |
| Gender | −0.220 | 0.551 | 0.803 | 0.391-1.651 |
| Primary/Recurrence | −1.042 | **0.009** | 0.353 | 0.161-0.774 |

bioinformatic analysis results showed that FBXO39 expression in GBM was significantly correlated with the survival of these patients. However, in our local cohort, the difference between high and low FBXO39 gene expression GBM survival was not significant (P = 0.12; Fig 6a). The discrepancy may be because of the small sample size. In this regard, it may be necessary to increase the number of cohort participants in future validation studies.

Previous studies have confirmed that CEP55 expression could distinguish cancers from control tissues in 21 cancer types [15], including hepatocellular [16], breast [17], liver [18], colorectal [19], lung cancer [20], etc., but CEP55 expression is not well documented in glioma, especially GBM [15,21–24]. A previously published study reported that CEP55 is highly expressed in the glioma tissue and that overexpression of CEP55 promoted glioma cells' proliferation and inhibited its apoptosis by activating the PI3K/Akt pathway [23]. This work demonstrated that overexpression of CEP55 might enhance the invasion and migration in the glioma tissue [22]. In our study, the level of CEP55 expression was significantly related to the survival of GBM patients following the analysis of the CEP55 gene expression and survival in 249 patients from the CGGA database. These results implied that CEP55 may play an important role in the proliferation or invasion of

GBM cells. However, the TCGA database analyses and our PCR results showed that CEP55 expression in GBM was not significantly correlated with the survival of GBM patients. The discrepancy may be because of the different samples (e.g., race, gene background, etc.), and also the number of sample tests.

Due to the limited expression of CTAs in tumors, and their high antigenicity, vaccines targeting CTAs such as MAGE-A3 and NY-ESO-1, have been used for developing novel therapeutics against several diseases [25]. Despite promising preclinical studies in several tumor types, however, the antitumor value of CTA-based vaccines has been restricted in clinical trials [25]. Considering the difference of GBM from other tumors, there are still many limitations and difficulties in the application of CTAs in the treatment of GBM patients. This is mainly because of the unclear expression profile of most CTA in GBM [26]. However, targeting the FBXO39 and CEP55 genes may contribute to developing novel GBM-specific biomarkers in future clinical trials and pre-clinical research. A study from the TCGA database provided a comprehensive evaluation of cancer-testis antigen CEP55 expression in various cancers such as GBM. Increased expression of CEP55 showed significant correlations with poor or better OS rates in various cancers, however, it was not significant in GBM patients. They reported that CEP55 was a potential biomarker and was correlated with immune infiltration and immuno-therapy efficacy in a pan-cancer study [27]. Unlike their study, in our study, CEP55 expression is significantly related to the survival outcomes in GBM patients that were observed in the CGGA database. This difference may be due to the use of two different datasets obtained by independent research groups. Overall, large preclinical and clinical studies are needed to assess targeting for FBXO39 and CEP55 in GBM patients.

This study identified an interesting finding. If explored further, the current work has strong potential to inform the neuro-oncology community. Specifically, the current data shows FBXO39 correlates with patient survival in the United States and CEP55 correlates with patient survival in China. Intriguingly, both these antigens appear to correlate with survival in the Middle Eastern cohort. Based on a multitude of factors (age, ethnicity, TCGA transcriptional subclass), it is difficult to interpret this observation but further analysis can illuminate a rationale for this observation [3]. Generating a cancer-testis signature by patient stratification, either by age at diagnosis or demographic, has tremendous potential as a biomarker. Nevertheless, trends in neuro-oncology stipulate that brain tumors are a unique entity unrelated to other cancers and tissues [28]. Considering this interpretation, cancer testis antigens may not be informative in the clinical diagnosis, or treatment, of GBM.

One might inquire about the significant genes obtained from this study. Certainly, various computer modeling algorithms and prediction methods have been applied to predict outcomes in medical research. Selecting the best model is contro-versial and is dependent on the interest point of the research, and research in this field continues [29]. The most important variables will be changed by changing the algorithm structure [29]. Multivariable Cox regression analysis identified the LEMD1 gene and primary/recurrence status as independent prognostic factors for OS in the CGAA cohort. In contrast, statistical significance was not observed in variables in the TCGA dataset (Tables 3 and 4). On the other hand, as we all know, the TCGA and the CGGA databases are the world's largest and most comprehensive gene expression public databases in GBM patients. But, gene expression of CTCFL, POTEG, and PRAME was not reported in the CGGA dataset (Table 4; S6 File). Moreover, the differences seen between the two databases can be due to the differences in genetics between the different populations. In this study, the significant genes were selected based on two different methods, therefore, the most important genes are different [29,30]. However, according to our study methods, CEP55 and FBXO39 genes were considered.

There are some limitations in our work. Firstly, although we confirmed the results of this work by bioinformatic analysis, additional tests, and clinical investigations are needed to find the precise function of FBXO39 and CEP55 in carcinogen-esis and development. Secondly, the prognostic value of the study would be improved with the inclusion of lower-grade glioma data. However, low-grade glioma was not analyzed separately in the study due to the small sample size in our cohort. Thirdly, it is recommended that genes (CTCFL, PLAC1, LEMD1, PRAME, POTEG, DKKL1, PBK, ATAD2, and TTK) can be tested using the same PCR system with a larger sample size. Fourthly, we could not identify antibodies

already used in the pathology laboratories and directly usable in immunohistochemistry. However, we have this in mind, and at the first opportunity, we would test anti-FBXO39 and CEP55 antibodies if a reliable source is identified. Fifthly, our study only analyzed CTAs and did not consider other genetic alterations, which also play an essential role in GBM biology. Taken together, future studies are needed to clarify the biological roles of FBXO39 and CEP55 to develop personalized treatments for GBM patients.

## Conclusion

The expression of CTAs of FBXO39 and CEP55 was high in GBM patients. FBXO39 and CEP55 can be considered important markers in determining the survival of GBM patients. However, further study is needed with a larger sample size to validate these findings and other cancer-testis antigens in correlation with survival in GBM patients. It is important to understand that GBMs differ both biologically and clinically from other tumors. Hence, survival outcomes assessment of GBM needs to be approached from different angles given its heterogeneity, genomic and epigenetic state, localization, race, tumor size, age of diagnosis, etc.

## Supporting information

**S1 File. RNA-seq data analysis from the TCGA database demonstrated that 2463 genes were significantly overexpressed and 3706 were significantly lower expressed in GBM tissues compared to normal samples.**
(XLSX)

**S2 File. Based on the RNA-seq data analysis from the CGGA database between GBM and normal samples, we obtained 6606 downregulated DEGs and 6249 upregulated DEGs.**
(XLSX)

**S3 File. The intersection between 279 CTAs in the CD database and the up and down-regulated genes of both other databases found 11 and, 8 CTAs, respectively.**
(XLSX)

**S4 File. The expressions of FBXO39 and CEP55 were significantly up-regulated in GBM tissues compared to normal tissues in our cohort.**
(XLSX)

**S5 File. Gene expression and related demographics data from the TCGA database performing Cox Regression in Excel.**
(XLSX)

**S6 File. Gene expression and related demographics data from the TCGA database performing Cox Regression in Excel.**
(XLSX)

## Acknowledgments

We would like to thank the National Institute for Medical Research Development (NIMAD) for their support throughout the research process.

## Author contributions

**Formal analysis:** Maryam Bazrgar, Taravat Yazdanian.

**Methodology:** Taravat Yazdanian, Abolhassan Ahmadiani.

**Project administration:** Parisa Azimi, Abolhassan Ahmadiani.

**Software:** Maryam Bazrgar, Taravat Yazdanian.

**Supervision:** Parisa Azimi, Mehdi Totonchi, Abolhassan Ahmadiani.

**Validation:** Maryam Bazrgar.

**Visualization:** Mehdi Totonchi, Abolhassan Ahmadiani.

**Writing – original draft:** Maryam Bazrgar, Taravat Yazdanian, Mehdi Totonchi, Abolhassan Ahmadiani.

**Writing – review & editing:** Maryam Bazrgar, Taravat Yazdanian, Mehdi Totonchi, Abolhassan Ahmadiani.

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
