## [Decision Letter · Decision Letter 0]

20 May 2024

Dear Dr. Azimi,

We look forward to receiving your revised manuscript.

Kind regards,

Girijesh Kumar Patel, PhD

Academic Editor

PLOS ONE

Journal Requirements:

Reviewers' comments:

Reviewer's Responses to Questions

**Comments to the Author**

1. Is the manuscript technically sound, and do the data support the conclusions?

Reviewer #1: Partly

Reviewer #2: Partly

2. Has the statistical analysis been performed appropriately and rigorously?

Reviewer #1: Yes

Reviewer #2: Yes

3. Have the authors made all data underlying the findings in their manuscript fully available?

Reviewer #1: Yes

Reviewer #2: Yes

4. Is the manuscript presented in an intelligible fashion and written in standard English?

Reviewer #1: Yes

Reviewer #2: No

**Reviewer #1** : This article focuses on glioblastoma multiform, a type of tumor with poor prognosis, and seeks potential diagnostic and therapeutic biomarkers related to prognosis, which has certain clinical significance. The language of the article is clear, and the results are reliably substantiated.

However, there are still some shortcomings in the article:

1.The article only conducts survival-related analysis on genes that are highly expressed in tumor tissues. May it be possible to further search for other meaningful genes among significantly down-regulated genes.

2.The article finds that the expression levels of FBXO39 and CEP55 are related to prognosis in the analysis of public data, but in their own cohort, the P-values for both are greater than 0.05. It may be beneficial to increase the number of cohort for further validation.

3.The article only validates the high expression of FBXO39 and CEP55 at the RNA level in tumor tissues, lacking protein-level verification. It is suggested to supplement with immunohistochemical staining of paraffin sections.

**Reviewer #2:**  Biomarker discovery in neuro-oncology has tremendous significance towards application in future therapies. Azimi et al. propose the utilization of cancer testis antigens as prospective biomarker candidates. Initially, the authors screen available RNA sequencing datasets from the United States and China. Specifically, the authors assess differentially expressed genes in glioblastoma (GBM) compared to normal brain. The authors then identified cancer testis genes that were specifically upregulated in the GBM cohorts and demonstrated correlation to patient survival. Following this discovery, Azimi et al. employed a prospective study of GBM patients, from their own institute, to validate a hypothesis that the expression of two testis antigens (FBXO39 and CEP55) correlates with patient survival. The manuscript is composed of two tables highlighting relevant clinical information and five figures. The main manuscript is supported by four spreadsheets featuring raw count data and statistics. The authors conclude that FBXO39 and CEP55 deserve consideration as biomarkers for GBM. While the prospective study is a unique dataset with substantial interest to the scientific community, the results underlying the authors' conclusion are exclusively correlative and lack necessary scientific rigor. Thus, the results, as currently written, do not merit the stated conclusion. Additionally, the written manuscript has components that may benefit from language editing. For these reasons, my recommendation would be to reject this manuscript.

The following are suggested minor revisions for the manuscript:

1. In paragraph 1 of the introduction, a comparison of patient characteristics and demographics from around the world would be beneficial. An example of one of these repositories is the US CBTRUS.

2. In the methods sections, please provide details on how normal brain tissue data was retrieved from TCGA and CGGA, including the number of normal specimens.

3. In the statistical analysis subsection of methods, please describe, in detail, the stratification of high and low expression samples used in the PCR and subsequent Kaplan Meier analysis.

4. Related to manuscript editing, in the "gene expression and overall survival analysis in our cohort" subsection, the language is misleading. In earlier sections, correlation was interpreted with a poor survival phenotype. However, in this section, the correlation refers to prolonged survival. This interpretation is valid but misleading. Additionally, the discussion section refers to the lack of correlation for some targets without presenting the necessary figure.

5. For figures 2A/B (and 4 A/B), it may be beneficial to provide the actual p-value.

The following are suggested major revisions for the manuscript:

1. Even though it was mentioned in the text that the survival trend was insignificant, the manuscript would be improved by presenting Kaplan Meier survival data for a larger panel of testis antigens. Presenting this negative data for both US and Chinese cohorts may be enlightening towards a personalized medicine approach for other patient cohorts.

2. For interpretation of survival data, multivariate analysis is required. First, the CGGA dataset includes primary and recurrent GBM cases. Second, there is an age disparity that may confound the results. Logistic regression or other modeling should be employed.

3. Both TCGA and CGGA datasets have low grade glioma information; the prospective study also has samples unused presumably based on a lower diagnosis grade. The prognostic value of the study would be improved with the inclusion of lower grade glioma data.

4. The utilization of this study to further assessment of these antigens as predictive markers in immunotherapy was mentioned in the introduction and discussion. Based on the scope of the project, if the authors do not want to assess patients treated with immunotherapy or assess immunotherapy targets pre-clinically, the innovation associated with predictive power in immunotherapy should be de-emphasized.

5. Similar to major revision #1, a larger panel of cancer testis antigens should be tested using the same PCR system.

The authors' current conclusion is misleading without further rigorous validation. However, the authors identified an interesting finding. If explored further, the current work has strong potential to inform the neuro-oncology community. Specifically, the current data shows FBXO39 correlates with patient survival in the United States and CEP55 correlates with patient survival in China. Intriguingly, both these antigens appears to correlate with survival in the Middle Eastern cohort. Based on a multitude of factors (age, ethnicity, TCGA transcriptional subclass), it is difficult to interpret this observation but further analysis can illuminate a rationale for this observation. Generating a cancer testis signature by patient stratification, either by age at diagnosis or demographic, has tremendous potential as a biomarker. Nevertheless, trends in neuro-oncology stipulate that brain tumors are a unique entity (dot: 10.1158/2159-8290.CD-23-1498), unrelated to other cancers and tissues. Considering this interpretation, cancer testis antigens may not be informative in the clinical diagnosis, or treatment, of GBM.

**Do you want your identity to be public for this peer review?** For information about this choice, including consent withdrawal, please see our Privacy Policy

Reviewer #1: No

Reviewer #2: **Yes: ** Artem D. Berezovsky

---

## [Author Response · Author response to Decision Letter 1]

28 May 2024

27 May 2024

Dear Editorial Committee,

MS: No. PONE-D-24-09413

Cancer/testis antigens FBXO39 and CEP55 expression correlates with survival in GBM patients

Thank you for your e-mail. Please find the following point-by-point responses as requested:

Comments to the Author

Reviewer #1:

This article focuses on glioblastoma multiform, a type of tumor with poor prognosis, and seeks potential diagnostic and therapeutic biomarkers related to prognosis, which has certain clinical significance. The language of the article is clear, and the results are reliably substantiated.

Thank you.

However, there are still some shortcomings in the article:

1.The article only conducts survival-related analysis on genes that are highly expressed in tumor tissues. May it be possible to further search for other meaningful genes among significantly down-regulated genes.

We thank the referee for this very interesting suggestion.

This was revised as suggested.

a. The following sentence was added to the Results:

“The intersection between 279 CTA and the down-regulated genes of both TCGA and CGGA databases showed that 8 CTAs are decreased in GBM compared to healthy people, including SPAG6, CABYR, GPAT2, ELOVL4, TMEFF1, TMEFF2, MAGEC3, and SYCE1 (Figure 1b). “ and “ However, no significant effect was indicated for the remaining down-regulated CTAs regarding OS.”

b. The following sentence was added to the Discussion:

“Moreover, indeed, by nature, the CTA-encoding genes should be found only within the genes that are activated in cancer, since by definition, they are only expressed in the testis and not in normal somatic tissues. However, the brain is a particular tissue concerning the CTA encoding gene expression, since many testis-specific genes are known to be also expressed in the brain. We have also considered the list of significantly down-regulated genes compared to normal brain samples and checked if any of them is present in the list of CTA-encoding genes. We then considered the association between their down-regulation and the patient's survival. However, no statistically significant effect was observed on down-regulated CTAs regarding OS, albeit, more studies are needed to confirm the findings.“

2.The article finds that the expression levels of FBXO39 and CEP55 are related to prognosis in the analysis of public data, but in their own cohort, the P-values for both are greater than 0.05. It may be beneficial to increase the number of cohort for further validation.

The following sentence was added to the Discussion: “In this regard, it may be necessary to increase the number of cohort participants in future validation studies.”

3. The article only validates the high expression of FBXO39 and CEP55 at the RNA level in tumor tissues, lacking protein-level verification. It is suggested to supplement with immunohistochemical staining of paraffin sections.

We thank the referee for these suggestions, however, although companies are commercializing these antibodies, we could not identify antibodies already used in the pathology laboratories and directly usable in immunohistochemistry. However, we have this in mind, and at the first opportunity, we would test anti-FBXO39 and CEP55 antibodies if a reliable source is identified.

Reviewer #2:

Biomarker discovery in neuro-oncology has tremendous significance towards application in future therapies. Azimi et al. propose the utilization of cancer testis antigens as prospective biomarker candidates. Initially, the authors screen available RNA sequencing datasets from the United States and China. Specifically, the authors assess differentially expressed genes in glioblastoma (GBM) compared to normal brain. The authors then identified cancer testis genes that were specifically upregulated in the GBM cohorts and demonstrated correlation to patient survival. Following this discovery, Azimi et al. employed a prospective study of GBM patients, from their own institute, to validate a hypothesis that the expression of two testis antigens (FBXO39 and CEP55) correlates with patient survival. The manuscript is composed of two tables highlighting relevant clinical information and five figures. The main manuscript is supported by four spreadsheets featuring raw count data and statistics. The authors conclude that FBXO39 and CEP55 deserve consideration as biomarkers for GBM. While the prospective study is a unique dataset with substantial interest to the scientific community, the results underlying the authors' conclusion are exclusively correlative and lack necessary scientific rigor. Thus, the results, as currently written, do not merit the stated conclusion. Additionally, the written manuscript has components that may benefit from language editing. For these reasons, my recommendation would be to reject this manuscript.

The following are suggested minor revisions for the manuscript:

1. In paragraph 1 of the introduction, a comparison of patient characteristics and demographics from around the world would be beneficial. An example of one of these repositories is the US CBTRUS.

The following sentence was added to the Introduction: “Glioblastomas accounted for 60.2% of all gliomas, 14.2 % of all CNS tumors, and 50.9% of all malignant CNS tumors. Incidence of GBM increased slightly statistically significantly from 2000-2004 and 2004-2019. With an incidence rate of 3.27 per 100,000 persons in the United States, and more common in males, it is uncommon in children and increases with age [2]. The diverse factors such as incidence rates, tumor location, age of diagnosis, race, and the role of molecular genetics in the diagnosis of GBM, etc., are reported in various regions of the world. However, making comparisons of some data e.g., GBM incidence was complex due to significant differences in the methodology of epidemiological reports [3] “

2. Ostrom QT, Price M, Neff C, Cioffi G, Waite KA, Kruchko C, et al. CBTRUS Statistical Report: Primary Brain and Other Central Nervous System Tumors Diagnosed in the United States in 2016-2020. Neuro Oncol. 2023 Oct 4;25(12 Suppl 2):iv1-iv99.

3. Grochans S, Cybulska AM, Simińska D, Korbecki J, Kojder K, Chlubek D, Baranowska-Bosiacka I. Epidemiology of Glioblastoma Multiforme-Literature Review. Cancers (Basel). 2022;14(10):2412.

2. In the methods sections, please provide details on how normal brain tissue data was retrieved from TCGA and CGGA, including the number of normal specimens

This was revised as suggested in the Method: “ The RNA-seq and relevant clinical data of GBM patients, and normal controls were acquired from The Cancer Genome Atlas (TCGA, https://www.cancer.gov/) and the Chinese Glioma Genome Atlas (CGGA; http://www.cgga.org.cn/) database (CGGA.mRNAseq_693) [10]. A total of 418 GBM and 25 normal brain samples, including 169 and 5 from TCGA and 249 and 20 from the CGGA were included in the present study.“

3. In the statistical analysis subsection of methods, please describe, in detail, the stratification of high and low expression samples used in the PCR and subsequent Kaplan Meier analysis.

The following sentence was added to Methods:

“For expression and survival analysis, we employed the median gene value to categorize tumor samples into high and low-expression groups. Subsequently, the expression data were merged with clinical survival data using sample barcodes. GraphPad Prism was employed to model these two groups' survival time and status. Log-rank test was utilized for the comparison of Survival Curves. “

4. Related to manuscript editing, in the "gene expression and overall survival analysis in our cohort" subsection, the language is misleading. In earlier sections, correlation was interpreted with a poor survival phenotype. However, in this section, the correlation refers to prolonged survival. This interpretation is valid but misleading.

This was revised as suggested.

Additionally, the discussion section refers to the lack of correlation for some targets without presenting the necessary figure.

The following sentence was added to the Results:

“As shown in Figure 4, genes (CTCFL, PLAC1, LEMD1, PRAME, CEP55, DKKL1, PBK, ATAD2, and TTK) from the GEPIA2 dataset, and genes (PLAC1, FBXO39, LEMD1, DKKL1, PBK, ATAD2, and TTK) of primary and recurrent GBM from the mRNA_seq693 of the CCGA cohort were not associated significantly with overall survival in GBM patients. For gene (POTEGE), and genes (CTCFL, PRAME, and POTEG), Kaplan-Meier was not analyzed due to the small sample size in the GEPIA2 and CGGA datasets, respectively. “

Figure 4. (a–p) Kaplan-Meier analysis of overall survival for GBM patients in the GEPIA2 using the TCGA cohort (a. CTCFL, b. PLAC1, c. LEMD1, d. PRAME, e. CEP55, f. DKKL1, g. PBK, h. ATAD2, and i. TTK), and primary GBM from the mRNA_seq693 of the CCGA (j. PLAC1, k. FBXO39, l. LEMD1, m. DKKL1, n. PBK, o. ATAD2, and p. TTK) based on low- and high-expression of genes. The red line represents samples with high expression of the genes, and the blue line represents the samples with low expression of genes. Among 11 genes, p < 0.05 was considered to be statistically different.

5. For figures 2A/B (and 4 A/B), it may be beneficial to provide the actual p-value.

This was revised as suggested.

The following are suggested major revisions for the manuscript:

1. Even though it was mentioned in the text that the survival trend was insignificant, the manuscript would be improved by presenting Kaplan Meier survival data for a larger panel of testis antigens. Presenting this negative data for both US and Chinese cohorts may be enlightening towards a personalized medicine approach for other patient cohorts.

The following sentence was added to the Results:

“As shown in Figure 4, genes (CTCFL, PLAC1, LEMD1, PRAME, CEP55, DKKL1, PBK, ATAD2, and TTK) from the GEPIA2 dataset, and genes (PLAC1, FBXO39, LEMD1, DKKL1, PBK, ATAD2, and TTK) of primary and recurrent GBM from the mRNA_seq693 of the CCGA cohort were not associated significantly with overall survival in GBM patients. For gene (POTEGE), and genes (CTCFL, PRAME, and POTEG), Kaplan-Meier was not analyzed due to the small sample size in the GEPIA2 and CGGA datasets, respectively. “

Figure 4. (a–p) Kaplan-Meier analysis of overall survival for GBM patients in the GEPIA2 using the TCGA cohort (a. CTCFL, b. PLAC1, c. LEMD1, d. PRAME, e. CEP55, f. DKKL1, g. PBK, h. ATAD2, and i. TTK), and primary GBM from the mRNA_seq693 of the CCGA (j. PLAC1, k. FBXO39, l. LEMD1, m. DKKL1, n. PBK, o. ATAD2, and p. TTK) based on low- and high-expression of genes. The red line represents samples with high expression of the genes, and the blue line represents the samples with low expression of genes. Among 11 genes, p < 0.05 was considered to be statistically different.

2. For the interpretation of survival data, multivariate analysis is required. First, the CGGA dataset includes primary and recurrent GBM cases. Second, there is an age disparity that may confound the results. Logistic regression or other modeling should be employed.

The following sentence was added to the Methods:

“Cox regression analysis of factors contributing to overall survival of GBM patients

Overall survival was defined as the time from the start of treatment to death or last follow-up based on public databases. This overall survival time was the outcome of interest. The prognostic prediction ability of 11 gene expression, age, gender, and primary/recurrence status of GBM patients was examined using multivariate Cox regression based on samples from the TCGA and CGGA databases, separately. Hazard ratios (HRs) and 95% confidence intervals (CIs) were calculated accordingly. Then, factors with significant prognostic values were assessed. “ and “The multivariate Cox regression analyses were employed to recognize the independent clinical prognostic factors using the Cox regression in SPSS. “

The following sentence was added to the Results:

“The multivariate Cox regression analysis was performed to obtain the independent prognostic feature factors. As shown in Table 3-4 ( Supplementary files 5-6), factors were found to be independently correlated to OS prognosis in the TCGA and the CGGA datasets, respectively. However, only, the LEMD1 gene and primary/recurrence status were significant in the CGAA cohort.“

The following sentence was added to the Discussion:

“ One might inquire about the significant genes obtained from this study. Certainly, various computer modeling algorithms and prediction methods have been applied to predict outcomes in medical research. Selecting the best model is controversial and is dependent on the interest point of the research, and research in this field continues [29]. The most important variables will be changed by changing the algorithm structure [29]. Multivariable Cox regression analysis identified the LEMD1 gene and primary/recurrence status as independent prognostic factors for OS in the CGAA cohort. In contrast, statistical significance was not observed in variables in the TCGA dataset (Tables 3 and 4). On the other hand, as we all know, the TCGA and the CGGA databases are the world's largest and most comprehensive gene expression public databases in GBM patients. But, gene expression of CTCFL, POTEG, and PRAME was not reported in the CGGA dataset (Table 4; Supplementary 6). Moreover, the differences seen between the two databases can be due to the differences in genetics between the different populations. In this study, the significant genes were selected based on two different methods, therefore, the most important genes are different [29-30]. However, according to our study methods, CEP55 and FBXO39 genes were considered. “

29. Kimberley D. Brosofske, Robert E. Froese, Michael J. Falkowski, et al. A Review of Methods for Mapping and Prediction of Inventory Attributes for Operational Forest Management. Forest Science. 2014; 60(4): 733–756.

30. Fury W, Batliwalla F, Gregersen PK, Li W: Overlapping probabilities of top ranking gene lists, hypergeometric distribution, and stringency of gene selection criterion. Engineering in Medicine and Biology Society, 2006. EMBS'06. 28th Annual International Conference of the IEEE. 2006, IEEE, 5531-5534.

3. Both TCGA and CGGA datasets have low grade glioma information; the prospective study also has samples unused presumably based on a lower diagnosis grade. The prognostic value of the study would be improved with the inclusion of lower grade glioma data.

a. We thank the referee for these suggestions, however, the aim of this study in phase 1 is to identify and validate the key CTA genes, whose expression correlates with the survival of GBM patients. However, we have this in mind, and at the first opportunity, we will perform a phase 2 study about low-grade glioma.

b. The following sentence was added to the Discussion: “Secondly, the prognostic value of the study would be improved with the inclusion of lower-grade glioma data. However, low-grade glioma was not analyzed separately in the study due to the small sample size in our cohort.”

4. The utilization of this study to further assessment of these antigens as predictive markers in immunotherapy was mentioned in the introduction and discussion. Based on the scope of the project, if the authors do not want to assess patients treated with immunotherapy or assess immunotherapy targets pre-clinically, the innovation associated with predictive power in immunotherapy should be de-emphasized.

a. The following sentence was added to the Introduction: “ In addition, it is also possible that these specific antigens may be potential targets for immunotherapy in future clinical trials and pre-clinical research. “

b. The following sentence was added to the Discussion:

“Despite promising preclinical studies in several tumor types, however, the antitumor value of CTA‑based vaccines has been restricted in clinical trials [25]. Considering the difference of GBM from other tumors, there are still many limitations and difficulties in the application of CTAs in the treatment of GBM patients. This is mainly because of the unclear expression profile of most CTA in GBM [26]. However, targeting the FBXO39 and CEP55 genes may contribute to developing novel GBM-specific biomarkers in future clinical trials and pre-clinical research. “

26. Li F, Liu C, Nong W, Li

---

## [Decision Letter · Decision Letter 1]

1 Aug 2024

Dear Dr. Azimi,

Thank you for submitting your manuscript to PLOS ONE. After careful consideration, we feel that it has merit but does not fully meet PLOS ONE’s publication criteria as it currently stands. Therefore, we invite you to submit a revised version (minor) of the manuscript that addresses the points raised during the review process.

We look forward to receiving your revised manuscript.

Kind regards,

Girijesh Kumar Patel, PhD

Academic Editor

PLOS ONE

Journal Requirements:

Reviewers' comments:

Reviewer's Responses to Questions

**Comments to the Author**

Reviewer #1: All comments have been addressed

Reviewer #3: All comments have been addressed

2. Is the manuscript technically sound, and do the data support the conclusions?

Reviewer #1: Yes

Reviewer #3: Yes

3. Has the statistical analysis been performed appropriately and rigorously?

Reviewer #1: Yes

Reviewer #3: I Don't Know

4. Have the authors made all data underlying the findings in their manuscript fully available?

Reviewer #1: Yes

Reviewer #3: Yes

5. Is the manuscript presented in an intelligible fashion and written in standard English?

Reviewer #1: Yes

Reviewer #3: Yes

**Reviewer #1:**  The author has proposed good improvement plans and corresponding explanations for the raised issues. No further concerns are related.

**Reviewer #3: ** This study explores the potential of FBXO39 and CEP55 as prognostic biomarkers for GBM. Utilizing RNA-seq data from the TCGA and CGGA databases, the study identifies differentially expressed genes and specifically highlights up-regulated cancer/testis antigens (CTAs) that correlate with overall survival in GBM patients. The findings are validated with an independent cohort of 29 GBM patients using RT-qPCR. The study concludes that high expression levels of FBXO39 and CEP55 are significantly associated with poorer overall survival, suggesting these genes as potential targets for prognostic assessment and therapeutic intervention in GBM. Further research with larger sample sizes and protein-level validation is recommended to strengthen these findings.

Title:

The title is clear and accurately reflects the content of the manuscript.

Abstract:

The abstract provides a good summary of the study's background, methods, results, and conclusions.

The mention of both databases (TCGA and CGGA) and the independent cohort is good, providing a clear understanding of the study's scope.

Consider revising sentences for conciseness and clarity.

Introduction

The introduction provides a clear background on GBM and the significance of cancer/testis antigens (CTAs) as biomarkers.

The inclusion of statistics and references to WHO classification and incidence rates enhances the context.

The rationale for studying FBXO39 and CEP55 is well-explained, linking it to the potential for novel therapeutic targets.

Methods

The methods section is detailed and logically structured.

Descriptions of data acquisition, DEG screening, and identification of CTAs are clear.

The explanation of survival analysis and the use of independent cohorts are well-detailed.

The addition of the Cox regression analysis for overall survival is a good inclusion, addressing one of the reviewer’s concerns.

Consider breaking down long paragraphs into smaller sections for better readability.

Results

The results are comprehensive and clearly presented.

Figures and tables support the textual data effectively.

Consider revising sentences for conciseness and clarity.

Discussion

The discussion contextualizes the findings within existing literature well.

The implications of FBXO39 and CEP55 as prognostic markers are clearly outlined.

Consider addressing the need for protein-level validation more directly, as it was a significant reviewer comment.

Conclusion

The conclusion summarizes the key findings and their potential implications succinctly.

The call for further studies with larger sample sizes is appropriate and necessary.

Overall Assessment

Clarity: The manuscript is generally clear, but some sentences can be revised for better readability and conciseness. Breaking down longer sections in the Methods and Results can also improve clarity.

Conciseness: The manuscript is concise overall but can benefit from removing redundant phrases and combining shorter sentences where possible.

Figures and Tables: Effective use of visual aids supports the findings.

References: The references are comprehensive and relevant.

Based on the clarity and conciseness, as well as the significant revisions made in response to reviewers' comments, this manuscript is suitable for publication with minor revisions. Addressing the suggestions for improving readability and ensuring grammatical precision will further enhance the manuscript's quality.

**Do you want your identity to be public for this peer review?** For information about this choice, including consent withdrawal, please see our Privacy Policy

Reviewer #1: No

Reviewer #3: No

---

## [Author Response · Author response to Decision Letter 2]

3 Aug 2024

2 August 2024

Dear Editorial Committee,

MS: No. PONE-D-24-09413R1

Cancer/testis antigens FBXO39 and CEP55 expression correlates with survival in GBM patients

Thank you for your e-mail. Please find the following point-by-point responses as requested:

Comments to the Author

1. If the authors have adequately addressed your comments raised in a previous round of review and you feel that this manuscript is now acceptable for publication, you may indicate that here to bypass the “Comments to the Author” section, enter your conflict of interest statement in the “Confidential to Editor” section, and submit your "Accept" recommendation.

Reviewer #1: All comments have been addressed

Reviewer #3: All comments have been addressed

Thank you.

2. Is the manuscript technically sound, and do the data support the conclusions?

Reviewer #1: Yes

Reviewer #3: Yes

3. Has the statistical analysis been performed appropriately and rigorously?

Reviewer #1: Yes

Reviewer #3: I Don't Know

4. Have the authors made all data underlying the findings in their manuscript fully available?

Reviewer #1: Yes

Reviewer #3: Yes

5. Is the manuscript presented in an intelligible fashion and written in standard English?

Reviewer #1: Yes

Reviewer #3: Yes

6. Review Comments to the Author

Reviewer #1: The author has proposed good improvement plans and corresponding explanations for the raised issues. No further concerns are related.

Thank you.

Reviewer #3: This study explores the potential of FBXO39 and CEP55 as prognostic biomarkers for GBM. Utilizing RNA-seq data from the TCGA and CGGA databases, the study identifies differentially expressed genes and specifically highlights up-regulated cancer/testis antigens (CTAs) that correlate with overall survival in GBM patients. The findings are validated with an independent cohort of 29 GBM patients using RT-qPCR. The study concludes that high expression levels of FBXO39 and CEP55 are significantly associated with poorer overall survival, suggesting these genes as potential targets for prognostic assessment and therapeutic intervention in GBM. Further research with larger sample sizes and protein-level validation is recommended to strengthen these findings.

Title:

The title is clear and accurately reflects the content of the manuscript.

Abstract:

The abstract provides a good summary of the study's background, methods, results, and conclusions.

The mention of both databases (TCGA and CGGA) and the independent cohort is good, providing a clear understanding of the study's scope.

Consider revising sentences for conciseness and clarity.

This was revised as suggested.

Introduction

The introduction provides a clear background on GBM and the significance of cancer/testis antigens (CTAs) as biomarkers.

The inclusion of statistics and references to WHO classification and incidence rates enhances the context.

The following sentence was added to the Introduction”

“The classification of brain tumors has undergone several changes over time due to various factors such as genetic variations. The World Health Organization (WHO) has played a key role in the effort [1].”

The following sentence was revised as suggested:

“With an incidence rate of 3.27 per 100,000 persons in the United States, and more common in males, it is uncommon in children and increases with age [2].”

The rationale for studying FBXO39 and CEP55 is well-explained, linking it to the potential for novel therapeutic targets.

Methods

The methods section is detailed and logically structured.

Descriptions of data acquisition, DEG screening, and identification of CTAs are clear.

The explanation of survival analysis and the use of independent cohorts are well-detailed.

The addition of the Cox regression analysis for overall survival is a good inclusion, addressing one of the reviewer’s concerns.

Consider breaking down long paragraphs into smaller sections for better readability.

This was revised as suggested.

The following sentence “ The prognostic prediction ability of 11 gene expression, age, gender, and primary/recurrence status of GBM patients was examined using multivariate Cox regression based on samples from the TCGA and CGGA databases, separately. “ was changed to “Multivariate Cox regression included 11 gene expressions, age, gender, and primary/recurrence status of GBM patients. Samples from the TCGA and CGGA databases, separately, were used. “

Results

The results are comprehensive and clearly presented.

Figures and tables support the textual data effectively.

Consider revising sentences for conciseness and clarity.

This was revised as suggested.

Discussion

The discussion contextualizes the findings within existing literature well.

The implications of FBXO39 and CEP55 as prognostic markers are clearly outlined.

Consider addressing the need for protein-level validation more directly, as it was a significant reviewer comment.

The following sentence was added to the Discussion:

“We could not identify antibodies already used in the pathology laboratories and directly usable in immunohistochemistry. However, we have this in mind, and at the first opportunity, we would test anti-FBXO39 and CEP55 antibodies if a reliable source is identified.”

Conclusion

The conclusion summarizes the key findings and their potential implications succinctly.

The call for further studies with larger sample sizes is appropriate and necessary.

Overall Assessment

Clarity: The manuscript is generally clear, but some sentences can be revised for better readability and conciseness. Breaking down longer sections in the Methods and Results can also improve clarity.

This was revised as suggested.

Conciseness: The manuscript is concise overall but can benefit from removing redundant phrases and combining shorter sentences where possible.

This was revised as suggested.

Figures and Tables: Effective use of visual aids supports the findings.

References: The references are comprehensive and relevant.

Based on the clarity and conciseness, as well as the significant revisions made in response to reviewers' comments, this manuscript is suitable for publication with minor revisions. Addressing the suggestions for improving readability and ensuring grammatical precision will further enhance the manuscript's quality.

This was revised as suggested.

I hope you find the corrections satisfactory.

Wish you all the best.

Yours sincerely

Azimi, P.

---

## [Decision Letter · Decision Letter 2]

30 Oct 2024

Dear Dr. Azimi,

We look forward to receiving your revised manuscript.

Kind regards,

Girijesh Kumar Patel, PhD

Academic Editor

PLOS ONE

Journal Requirements:

Additional Editor Comments:

**Based on the reviewer's evaluation, please address the raised concerns.**

Reviewers' comments:

Reviewer's Responses to Questions

**Comments to the Author**

Reviewer #4: All comments have been addressed

2. Is the manuscript technically sound, and do the data support the conclusions?

Reviewer #4: Partly

3. Has the statistical analysis been performed appropriately and rigorously?

Reviewer #4: Yes

4. Have the authors made all data underlying the findings in their manuscript fully available?

Reviewer #4: Yes

5. Is the manuscript presented in an intelligible fashion and written in standard English?

Reviewer #4: Yes

Reviewer #4: Thank you for inviting me to review the article. In this article, the authors explored cancer testic antigen genes FBXO39 and CEP55 and their correlation with survival in GBM patients, using TCGA and CGGA databases and multiple bioinformatic methods. I generally think that the article matches the journal's scope and quality. However, there are several questions that I would like authors to address before I can be fully convinced.

1. Figure 2a and Figure 2b showed that both FBXO39 and CEP55 were significantly increased in TCGA data. However, there is huge gap in case numbers between normal and GBM patients, 5 of normal vs. 169 of GBM. Is this comparison trustable? Actually as I tried to analyze using TCGA-GTEx (there are 207 cases in GTEx group, representing normal brain tissue), there was no significance between TCGA and normal. Therefore, I would like authors to explain why they chose to use TCGA data only instead of including GTEx data to make the result of NORMAL more representative.

2. As I checked on GDC data portal, there are 617 cases in TCGA-GBM project. The authors used 169 cases from TCGA. Can authors explain their selection criteria of TCGA cases? How did they avoid selection bias?

**Do you want your identity to be public for this peer review?** For information about this choice, including consent withdrawal, please see our Privacy Policy

Reviewer #4: No

---

## [Author Response · Author response to Decision Letter 3]

10 Dec 2024

8 December 2024

Dear Editorial Committee,

MS: No. PONE-D-24-09413R2

Cancer/testis antigens FBXO39 and CEP55 expression correlates with survival in GBM patients

Thank you for your e-mail. Please find the following point-by-point responses as requested:

Comments to the Author

1. If the authors have adequately addressed your comments raised in a previous round of review and you feel that this manuscript is now acceptable for publication, you may indicate that here to bypass the “Comments to the Author” section, enter your conflict of interest statement in the “Confidential to Editor” section, and submit your "Accept" recommendation.

Reviewer #4: All comments have been addressed

Thank you.

2. Is the manuscript technically sound, and do the data support the conclusions?

Reviewer #4: Partly

3. Has the statistical analysis been performed appropriately and rigorously?

Reviewer #4: Yes

4. Have the authors made all data underlying the findings in their manuscript fully available?

Reviewer #4: Yes

5. Is the manuscript presented in an intelligible fashion and written in standard English?

Reviewer #4: Yes

6. Review Comments to the Author

Reviewer #4: Thank you for inviting me to review the article. In this article, the authors explored cancer testis antigen genes FBXO39 and CEP55 and their correlation with survival in GBM patients, using TCGA and CGGA databases and multiple bioinformatic methods. I generally think that the article matches the journal's scope and quality. However, there are several questions that I would like authors to address before I can be fully convinced.

1. Figure 2a and Figure 2b showed that both FBXO39 and CEP55 were significantly increased in TCGA data. However, there is a huge gap in case numbers between normal and GBM patients, 5 of normal vs. 169 of GBM. Is this comparison trustable? Actually as I tried to analyze using TCGA-GTEx (there are 207 cases in GTEx group, representing normal brain tissue), there was no significance between TCGA and normal. Therefore, I would like the authors to explain why they chose to use TCGA data only instead of including GTEx data to make the result of NORMAL more representative.

We appreciate the reviewer's insightful comments regarding the disparity in sample sizes between normal (5 samples) and GBM (169 samples) patients in the TCGA dataset.

Our decision to utilize TCGA data exclusively for this analysis was based on several factors:

Matched Samples: The normal samples from TCGA are derived from adjacent tissues of cancer patients, which may provide relevant biological context compared to completely healthy tissue samples. However, we acknowledge that these samples may carry some tumor microenvironment signals, potentially influencing gene expression levels.

Statistical Power: While the number of normal samples is limited, TCGA's extensive dataset for GBM provides a robust basis for statistical analysis, allowing us to identify significant differences in gene expression that may be clinically relevant.

We recognize that incorporating GTEx data could enhance the representativeness of normal brain tissue comparisons.

To address the reviewer's suggestion, we conducted additional analyses that include GTEx data.

According to the dear reviewer's opinion, we used the set of normal GTEx data and normal TCGA samples. The results of our analyzes using gepia2 (http://gepia2.cancer-pku.cn/#analysis) showed that the CEP55 expression in GBM samples is increased compared to normal tissue but the FBXO39 expression in GBM samples was not significantly different compared to normal tissue. (You can see the results below).

This discrepancy may arise from differences in sample processing and batch effects between the two datasets, as highlighted in existing literature 1, 2, 3.

1. https://www.nature.com/articles/sdata201861

(Unifying cancer and normal RNA sequencing data from different sources)

2. https://ucsc-xena.gitbook.io/project/how-do-i/tumor-vs-normal

(How do I compare tumor vs normal expression?)

3. https://www.biostars.org/p/9502850/

(Combining GTEx and TCGA data)

We appreciate the reviewer’s recommendation to include GTEx data for a more comprehensive analysis. This will not only strengthen our results but also provide a clearer understanding of gene expression dynamics in GBM relative to healthy brain tissue.

We hope this response clarifies our rationale for using TCGA data and outlines our commitment to enhancing the robustness of our findings through further analysis. Thank you for your valuable feedback.

FBXO39:

CEP55:

2. As I checked on GDC data portal, there are 617 cases in TCGA-GBM project. The authors used 169 cases from TCGA. Can authors explain their selection criteria of TCGA cases? How did they avoid selection bias?

We sincerely thank the reviewer for his thorough observations. While it is true that the GDC data portal lists a total of 617 cases, this figure encompasses all GBM samples with various analyses performed on these samples, including Copy Number Variation, miRNA-Seq, DNA Methylation, and RNA-Seq. we specifically focused on GBM RNA-seq data, along with 5 normal samples obtained from adjacent tissues of cancer patients. We have included the following command to facilitate the download of data related to the TCGA-GBM project.

We used the TCGAbiolinks R package to download GBM RNAseq data.

Query command = GDCquery (project = "TCGA-GBM", data.category = "Transcriptome Profiling" data.type = "Gene Expression Quantification",

workflow.type = "STAR - Counts",

legacy = F, experimental.strategy = "RNA-Seq")

7. PLOS authors have the option to publish the peer review history of their article (what does this mean?). If published, this will include your full peer review and any attached files.

Do you want your identity to be public for this peer review? For information about this choice, including consent withdrawal, please see our Privacy Policy.

Reviewer #4: No

I hope you find the corrections satisfactory.

Wish you all the best.

Yours sincerely

Azimi, P.

---

## [Editor Report · Decision Letter 3]

23 May 2025

Cancer/testis antigens FBXO39 and CEP55 expression correlates with survival in GBM patients

PONE-D-24-09413R3

Dear Dr. Azimi,

We’re pleased to inform you that your manuscript has been judged scientifically suitable for publication and will be formally accepted for publication once it meets all outstanding technical requirements.

Kind regards,

Qinghua Shi

Academic Editor

PLOS ONE

Additional Editor Comments (optional):

I have checked the authors' revised manuscript and point-to-point response to reviewers' comments to authors, and find that the authors have addressed all the concerns of the reviewers'. Thus, I now suggest that the manuscript can be accepted for publication.

However, the authors should also modify the abstract based on their responses to reviewers' comments.
---

## [Editor Report · Acceptance letter]

PONE-D-24-09413R3

PLOS ONE

Dear Dr. Azimi,

I'm pleased to inform you that your manuscript has been deemed suitable for publication in PLOS ONE. Congratulations! Your manuscript is now being handed over to our production team.

Kind regards,

on behalf of

Professor Qinghua Shi

Academic Editor

PLOS ONE